# Molecular Detection of Porcine Parvovirus 5 in Domestic Pigs in Russia and Propagation of Field Isolates in Primary Porcine Testicular Cells

**DOI:** 10.3390/vetsci12060535

**Published:** 2025-06-01

**Authors:** Afshona Anoyatbekova, Alina Komina, Natalia Vlasova, Ekaterina Kononova, Alexey Gulyukin, Nikita Krasnikov, Anton Yuzhakov

**Affiliations:** Federal State Budget Scientific Institution “Federal Scientific Center VIEV”, Moscow 109428, Russia

**Keywords:** parvoviruses, porcine parvovirus 5, phylogenetic analysis, cells, isolation, field isolates

## Abstract

Porcine parvovirus 5 (PPV5) was first detected in 2013 in the USA. Thereafter, virus circulation has been confirmed worldwide. This study considers the detection and molecular epidemiology of PPV5 in the domestic pig population in Russia and the first isolation of the virus in cell culture. PPV5 was common in the investigated pig farms in Russia, and the overall detection rate was 8.9%. The virus was identified in 11 pig farms out of the 20 investigated. The isolates demonstrated high nucleotide identities with the globally detected strains. Additionally, PPV5 was propagated in primary porcine testicular cells.

## 1. Introduction

Parvoviruses (PVs) are ubiquitous in nature and infect a diverse array of mammals, birds, and insects [1]. According to the International Committee on the Taxonomy of Viruses (*ICTV*), PVs are assigned to the *Parvoviridae* family, which consists of three subfamilies, namely the arthropod-infecting *Densovirinae*, the vertebrate-infecting *Parvovirinae*, and the *Hamaparvovirinae*, which infects both vertebrates and invertebrates [2].

In recent years, advances in sequencing technologies have led to the frequent detection of novel parvoviruses in animals. Within the swine population, eight porcine parvoviruses (PPVs), PPV1 through PPV8, have been identified [2]. PPVs are a group of comparatively small (25 nm in diameter), non-enveloped, single-stranded DNA (ssDNA) viruses with a genome of 4–6 kilobases (kb) in size that generally contains two open reading frames (ORFs), encoding for non-structural proteins (NSPs) and viral capsid proteins (VPs) [1,3].

Among PPVs, there are currently four recognized genera, which are *Protoparvovirus*, *Tetraparvovirus*, and *Copiparvovirus*, belonging to the *Parvovirinae* subfamily, and *Chaphamaparvovirus*, belonging to *Hamaparvovirinae*. In turn, seven genetically divergent species of PPVs within the genera have been classified as follows: PPV1 and PPV8 belong to *Protoparvovirus*, PPV2 and PPV3 to *Tetraparvovirus*, PPV4 and PPV6 to *Copiparvovirus*, and PPV7 to the *Chaphamaparvovirus* genus [2,4]. Only porcine parvovirus 5 (PPV5) remains unclassified so far. Although the genetic variability of all PPVs is well studied, their pathogenic exposure, with the exception of PPV1 and tentatively PPV2, is still ambiguous [5].

PPV5 was first identified in 2013 in the USA in lung samples of pigs [6,7]. The genome of PPV5 equals 5756–5805 nt and contains two major ORFs. ORF1 encodes 601 amino acids (aa) of NSP, and ORF2 encodes VP1 with 991 aa, respectively [6]. PPV5 was detected in various samples from healthy pigs and pigs with different syndromes, including porcine respiratory disease complex (PRDC) and porcine circovirus-associated disease (PCVAD) [8,9]. However, the association between PPV5 and these syndromes has not been defined yet. To date, PPV5 circulation has been reported in domestic pigs in China, Brazil, Mexico, Colombia, South Korea, Poland, and Italy and in wild boars in South Korea and Russia [8,10,11,12,13,14,15,16,17,18,19]. Notably, PPV5 was found not only in pigs on farms but also in pork products [10]. Despite the high prevalence of PPV5 in many countries, it remains debatable whether the virus causes disease in pigs. Furthermore, even though PPV5 has been identified for more than a decade, neither its biological characteristics nor its isolation in a cell culture system have been investigated. Consequently, the characterization of PPV5 in vitro is of scientific interest and will provide a basis for its study in vivo.

In this study, we present the first identification of PPV5 in domestic pigs in Russia and genetic characteristics of the virus. Additionally, we report on the propagation of the virus in primary cell culture.

## 2. Materials and Methods

### 2.1. Samples, DNA Extraction, and Quantitative Real-Time PCR (qPCR)

A total of 984 serum samples were collected from 20 farrow-to-finish pig farms located in 10 regions of Russia, namely the Vologda, Kemerovo, Moscow, Sverdlovsk, Tomsk, Belgorod, and Pskov Regions; the Krasnoyarsk Krai; and the Republics of Buryatia and Mordovia. Samples were collected between 2020 and 2023 as part of the routine diagnostic surveillance on farms, so the health status of animals was unknown. All samples were individually collected, transported in iceboxes, and stored at −70 °C for subsequent analysis.

The age category was known merely for 516 animals as per the accompanying documentation. Consequently, the samples were divided into three age groups, with sows (*n* = 92), weaned piglets (*n* = 94), and fattening pigs (*n* = 330).

Prior to nucleic acid extraction, serum samples were pooled by five. Total DNA was extracted from 100 µL of each serum pool using the “RIBO-prep” commercial kit (FBIS Central Research Institute of Epidemiology of Rospotrebnadzor, Moscow, Russia), following the manufacturer’s recommendations. The qPCR was carried out with primers and a probe targeting a 704 nt fragment of the NSP gene designed by Xiao et al. [6]. The 25.0 µL PCR reaction volume included 2.5 µL of extracted DNA, 2.5 µL of 10X Taq Buffer (Alpha-ferment, Moscow, Russia), 0.5 µL of dNTPs mix (New England Biolabs, Ipswich, MA, USA), 16.25 µL of nuclease-free water, 0.25 µL Taq Polymerase (Alpha-ferment, Moscow, Russia), 1.0 µL of 10 pM both forward and reverse primers, and 1.0 µL of 10 pM of the probe. The PCR amplification protocol was performed according to Komina et al. [18]. The positive pools were then tested individually. Samples with Ct ≤ 35 were considered positive.

### 2.2. Sanger Sequencing and Phylogenetic Analysis

For Sanger sequencing, samples with the lowest Ct values were selected, and the capsid protein gene was amplified with the previously designed primer sets [18]. The PCR protocol was performed according to Komina et al. [18]. The PCR products were analyzed by electrophoresis using 1% agarose gel containing a Tris–acetate buffer solution (pH 8.0) and ethidium bromide (0.5 µg/mL). The amplified fragments were purified from agarose gel using the Monarch DNA Gel Extraction Kit (New England Biolabs, Ipswich, MA, USA) according to the manufacturer’s instructions and further stored at −20 °C until subsequent analysis. Sanger sequencing was conducted in both directions using the Big Dye 3.1 Terminator Cycle Sequencing Kit (Thermo Fisher Scientific, Carlsbad, CA, USA) and carried out on the ABI PRISM 3130 Genetic Analyzer. The obtained sequence chromatograms were analyzed and assembled into consensus sequences using SeqMan Lasergene 11.1.0 software (DNASTAR, Madison, WI, USA).

A phylogenetic analysis was conducted based on the capsid protein gene. The sequences were aligned using the ClustalW algorithm incorporated into UGENE v52.0 software [20] and, subsequently, the multiple alignment was trimmed to the capsid protein-coding sequence level. The phylogenetic dendrogram was inferred by the maximum likelihood (ML) method with the GTR model supported by 1000 bootstrap replicates (MEGA 7.0) [21]. The genetic distances were calculated by the Kimura-2 parameter method.

### 2.3. Primary Porcine Testicular Cell (PPTC) Preparation

PPTCs were derived from testes of 3–5-week-old healthy piglets. The testes were obtained during castration under sterile conditions and transported in Eagle MEM Medium (PanEco, Moscow, Russia) containing 100 mg/mL penicillin and 100 mg/mL streptomycin (PanEco, Moscow, Russia) to the laboratory. Aseptically, the tunica albuginea of the testes was cut lengthwise, and the parenchyma was enucleated. The testicular parenchyma was sliced into tiny pieces and digested by trypsin (0.25%) at 37 °C for 30 min. PPTCs were prepared according to the described protocol [22]. For cell seeding, the pellet was diluted in Eagle MEM Medium (PanEco, Moscow, Russia) and Lactalbumin Hydrolysate Medium (PanEco, Moscow, Russia) in an equal volume, supplemented with 10% fetal bovine serum (FBS) (Biosera, Cholet, France), L-glutamine (PanEco, Moscow, Russia), and 100 mg/mL penicillin and streptomycin (PanEco, Moscow, Russia). The cells were seeded in a 2 × 10^6^ cells/mL concentration in a T-175 cm^2^ culture flask (Nest, Wuxi, China) and incubated at 37 °C.

After 24 h of seeding, the growth medium was replaced by the fresh medium. Once the PPTCs reached confluency (4 days post seeding), cells were dissociated from the cultural flasks by trypsin–EDTA (0.25%) solution (PanEco, Moscow, Russia) and centrifuged for 20 min at 2500× *g*. Then, the cell pellet was resuspended in a cryoprotectant consisting of 10% DMSO (Servicebio, Wuhan, China), 50% Eagle MEM Medium, and 40% fetal bovine serum (FBS) (Biosera, Cholet, France) and stored under liquid nitrogen (−196 °C) for 2–3 months. Prior to freeze-down and virus isolation, PPTCs were verified to be free from the spontaneous contamination with PPVs (PPV1–PPV8), pestiviruses (APPV, BVDV, CSFV), circoviruses (PCV1-3), PRRSV, and *Mycoplasma* spp. by PCR, as was described previously [18].

### 2.4. PPV5 Isolation in PPTCs

For PPV5 isolation, frozen PPTCs stored in liquid nitrogen were thawed and maintained in a growth medium in a 25 cm^2^ culture flask (FuDau, Luoyang, China) and were incubated in a humidified 37 °C and 5% CO_2_ incubator. After 24 h, the old growth medium was discarded from the flasks and fresh medium was added. Upon reaching confluency, cells were dispersed by trypsin–EDTA (0.25%) (PanEco, Moscow, Russia) and reseeded into the secondary cultures. Trypsin–EDTA was neutralized, and cells were resuspended in Eagle MEM Medium and Lactalbumin Hydrolysate Medium (1:1) containing 10% FBS and antibiotics (100 mg/mL penicillin and streptomycin) at a concentration of 1 × 10^6^ cells/mL. Three hours after seeding, when cells attached to the culture flasks, the growth medium was gently removed. Then, cells were inoculated with PPV5-positive serum samples at a dilution of 1:10 in a volume of 2 mL. The mock-infected cells were inoculated with Eagle MEM Medium by the same procedure. After 12 h of adsorption at 4 °C, growth medium in a volume of 8 mL was added to the flasks and incubated at 37 °C in a 5% CO_2_ atmosphere for 4–5 days. Daily microscopy was performed. After reaching confluency in mock-infected cells, cells were harvested by two freeze–thaw cycles (−70° and +8 °C). Finally, the PPV5 stock suspension was obtained, aliquoted, and stored at −70° or used directly for the next virus passage. The supernatants of mock-infected and PPV5-infected PPTCs were collected after each passage and tested for the presence of viral DNA by qPCR and spontaneous contamination with other viruses (see Section 2.3). Seven serial PPV5 passages were conducted in PPTCs.

### 2.5. Azure–Eosin (AE) Staining

For AE staining, we used cells infected with the Moscow-4060 strain at the level of the 6th passage. The infection method and cultivation conditions were the same as mentioned above. Cell cultures were infected on sterile coverslips in a 6-well plate (SPL, Pocheon, Republic of Korea) and were incubated at 37 °C with 5% CO_2_. Upon confluency, the growth medium was gently replaced. PPV5-infected and mock-infected coverslips were fixed using 96% ethanol and AE dye in a ratio of 15:1 and then stained with AE, as described previously [23]. Cells were viewed on the Zeiss Axio Scope 1 light microscope (Carl Zeiss Microscopy GmbH, Oberkochen, Germany). The ADF PRO 08 camera with a suitable microscope interface was used to obtain images.

### 2.6. Statistical Analysis

Statistical analysis was conducted using Past 4.17 software. Differences in the prevalence of PPV5 among different age groups were investigated using Fisher’s exact criterion by pairwise comparisons. Results with a *p*-value of <0.05 were considered statistically significant.

### 2.7. Ethical Statement

Sampling in this study conformed to the institutional guidelines approved by the Ethical and Animal Welfare Committee of the Federal State Budget Scientific Institution “Federal Scientific Center VIEV” (Moscow, Russia), approval number 662/22 from 17 June 2021. The study was conducted in accordance with the local legislation and institutional requirements.

## 3. Results

### 3.1. PPV5 Detection Rates in Domestic Pigs in Russian Pig Farms

In total, 984 samples obtained from 20 pig farms located in 10 regions of Russia were tested for PPV5 presence by qPCR. The overall positive detection rate of PPV5 in investigated farms from 2020 to 2023 was 8.9% (88/984). PPV5 was identified in 11 pig farms located in eight regions of Russia (Table 1, Appendix A).

The highest PPV5 detection rate was estimated at the farms of the Belgorod Region (17.1%), followed by the Sverdlovsk Region (15.9%), the Republic of Buryatia (14.7%), and the Moscow Region (14.0%). A lower percentage of virus detection rates was observed at the farms of the Krasnoyarsk Krai (10.8%), the Tomsk (8.4%), and Kemerovo (6.8%) Regions and the Republic of Mordovia (5.6%). All samples from the farms of the Vologda and Pskov Regions were negative for PPV5.

It should be noted that the number of obtained samples varied significantly in each investigated year. In 2020, samples were obtained only from the KK2 and KK4 farms of the Krasnoyarsk Krai, and all were negative for PPV5. In 2021, samples from the VR and MR1 farms of the Vologda and Moscow Regions were tested, and the overall PPV5 detection rate was 8.75% (7/80). The first PPV5-positive case was noted at the MR1 farm of the Moscow Region, with 16.6% (7/42) of positive samples.

In 2022, we received the largest number (*n* = 731) of serum samples from 16 farms located in eight regions, and the overall percentage of PPV5-positive samples was 11.1% (81/731). In the Republic of Buryatia, two pig farms out of three were PPV5-positive, RB2 and RB3, with the virus detection rate equal to 11.6% (5/43) and 22.9% (16/70), respectively. From the Moscow Region, a limited number (*n* = 8) of serum samples were collected only from the MR2 farm, and all were PPV5-negative. From the Kemerovo Region, a total of 59 samples from two pig farms, KR1 and KR2, were tested. PPV5 was detected in samples from the KR1 farms with a positive rate of 7.8% (4/51). Seven PPV5-positive cases were found in the SR1 farm from the Sverdlovsk Region. In the Tomsk Region, samples from two farms, TR1 and TR2, were positive with 2.1% (2/95) and 25.0% (9/36) detection rates, respectively. PPV5 was detected in two pig farms (KK2 and KK3) out of four in the Krasnoyarsk Krai, with a positive rate of 17.4% (12/69) and 30% (3/10) in KK2 and KK3, respectively. In the Republic of Mordovia, PPV5 was identified in 11.3% of samples collected from the RM2 farm.

In 2023, serum samples were obtained from two farms located in the Pskov Region, but the virus was not detected in any of the tested samples.

### 3.2. PPV5 Detection Rates in Pigs of Different Age Groups

In total, 516 samples obtained from domestic pigs were divided into three age categories, namely weaned, fattening pigs, and sows (Table 2).

PPV5 detection frequency was observed in fattening pigs and sows that was statistically significant (*p* < 0.01). The fattening pigs were the group with the highest PPV5 detection rate (13.6%), where the virus was detected in 45 samples out of 330. PPV5 in sows was detected in 13.0% (12/92) of cases. Weaned piglets (*n* = 94) were the group with the lowest virus detection rate (1.1%).

### 3.3. Phylogenetic Analysis of PPV5 Isolates

The samples with the lowest Ct values in qPCR were selected for capsid protein gene sequencing. In total, two complete (2976 nt) and three near-complete (2964–2970 nt) sequences were obtained (Appendix A). For phylogenetic analysis, 26 sequences of PPV5 isolates and strains were derived from the NCBI GenBank (Figure 1, Appendix A).

The phylogenetic tree based on the capsid protein gene demonstrated that two Russian isolates, Tomsk-84 and Buryatia-302, were positioned in a distinct cluster with the Colombian isolate PPV5/COL/Valle511/OR355614/2021 and carried 99.9% nucleotide identity. The Tomsk-74 isolate formed a monophyletic clade with the OK/USA/MW051673/2019 isolate from the USA. Two isolates from the Moscow Region, Moscow-4060 and Moscow-019, were identical to each other and clustered together with the Chinese isolate (AH-PPV520178-3/MW853953). The MosWB28/PQ757601/2023 isolate, previously detected in the wild boar in Russia, showed 99.1–99.3% nucleotide identity with isolates from domestic pigs but did not cluster with any of them. All the sequenced isolates had a high nucleotide identity among each other (99.1–100%) and with strains from the NCBI GenBank (98.7–99.6%).

### 3.4. PPV5 Isolation and Propagation in Cell Culture

PPV5-positive serum samples, Moscow-4060 and Tomsk-84, which were characterized by the lowest Ct values, with 13.9 and 16.3, respectively, were utilized for the virus isolation conducted in PPTCs. For each virus passage, cells from liquid nitrogen were revived. The virus was cultivated for 4–5 days until a complete monolayer formed in the mock-infected cells. The PPV5 replication was assessed by qPCR and monitored daily by microscopy. While cultivating Tomsk-84, we detected PPV5 DNA solely in two passages (Table 3).

The Ct value of the Tomsk-84 strain in P1 equaled 21.0 and in P2, it was 23.9. In P3, no viral DNA was detected, and further cultivation of this strain was discontinued. During Moscow-4060 cultivation, continuous propagation of the virus in PPTCs over the course of seven passages was noticed. Despite the ten-fold dilution of the inoculum with each serial passage, we observed the virus accumulation on the last passages with further decreases in Ct value.

At the time of the virus cultivation, we compared the alterations that occurred in infected and mock-infected cells by light microscopy (Figure 2).

In addition, the cytomorphological staining of cells was performed for a more accurate characterization of the PPV5-infected and mock-infected cell culture (Figure 3).

It was noticed that the monolayer of mock-infected cells was dense and consisted of mixed types of cells, with both epithelial- and fibroblast-like cells. The cells’ boundaries were clearly defined, and cells had different sizes. The cell nuclei were oval or round in shape and contained one to four nucleoli. The binucleated cells were found in the cell’s population. In the cytoplasm, fine granularity was observed (Figure 2A and Figure 3A,B).

The morphological changes in cells were noticed when infected with the Moscow-4060 strain (Figure 2B,C) on the fourth to fifth day post-infection. The monolayer of the infected cell culture was less dense (Figure 2B and Figure 3C,D). In some areas, it was unformed or destroyed. The rounding up and detachment of cells from the monolayer were noticed (Figure 2B). In the nuclei, pyknosis (nuclear shrinking), alteration, and violation of membrane integrity were observed (Figure 3H). It was noted that the cells in the infected culture increased greatly in size (Figure 2E,F and Figure 3D). In such cells, the multinucleation or symplast formation was observed (Figure 2E,F and Figure 3(C2),F,G). The cytoplasm of infected cells was highly vacuolated (Figure 2C,D and Figure 3(C1)). The large vacuoles were gathered around the nucleus, while small ones were seen in the cytoplasm (Figure 2D and Figure 3D). In some cases, due to the strong vacuolization in the cytoplasm, the shape of the nucleus was greatly altered (Figure 2C and Figure 3E,I). In such nuclei, the nucleoli were enlarged (Figure 3E,I). The number of altered cells in one field of view varied from 2 to 6% at a magnification of 200×. The degenerative alteration in the infected cells was noticed from the first passage of the Moscow-4060 strain.

## 4. Discussion

After being initially reported in the USA in 2013, PPV5 has been found in a number of countries globally [6,8,9,10,11,12,13,14,16,18]. This study aimed to investigate the PPV5 circulation in domestic pigs in Russia and to isolate the virus in cell culture. Retrospectively, 984 serum samples collected between 2020 and 2023 from 20 pig farms located in 10 regions of Russia were tested by qPCR for the presence of PPV5. Among all the farms, PPV5 circulation was established in 11 farms located in eight regions, and the overall PPV5 detection rate was equal to 8.9% (88/984). In comparison with other studies, positive rates of PPV5 in serum samples were recorded at 19.7% (50/254) in Poland, at 20.5% (48/234) in Colombia, and at 9.5% (25/268) in South Korea, respectively [8,13,16]. Alternatively, for PPV5 detection, tissue samples were taken frequently, and the virus prevalence in China was estimated at 9.19% (40/435) in various tissue samples [19]. Moreover, high detection rates in lung specimens, aborted fetuses, and feces were observed [8,13].

The first PPV5-positive case was identified in a farm (MR1) of the Moscow Region in 2021. Nevertheless, no samples from other farms were obtained during that period. The vast majority of the samples were collected from pig farms in 2022. The frequent detection of PPV5 was noticed in the farms of the Belgorod (17.1%), Sverdlovsk (15.9%), and Moscow (14.0%) Regions. The farms in the Krasnoyarsk Krai (10.8%), the Republic of Mordovia (5.6%), the Tomsk Region (8.4%), and Kemerovo Region (6.8%) demonstrated lower detection rates. Based on our findings, it follows that PPV5 was most often detected in regions of Russia where intensive pig farming is carried out. It should be noted that pig farming is practiced in nearly every region of Russia, albeit with varying intensity depending on the geographic conditions and economic circumstances.

According to numerous studies, PPV5 has been found in pigs of different ages, ranging from neonatal to adult groups [6,8,13]. Following this, there is no precise sampling procedure, likely due to the unknown pathogenesis of the virus and the age at which pigs are most susceptible. In our study, we were able to determine the age category only for 516 animals based on the data in companion documents. The PPV5 genome was detected in the serum samples of fattening pigs in 13.6% (45/330), in sows in 13.0% (12/92), and in weaned piglets in 1.1% (1/94) of cases. Similar differences in the detection rates of PPV5 in serum samples between various age groups of pigs were previously reported [6,8,13,19]. Moreover, in our study, fattening pigs were the age group with the highest PPV5 detection rate. These results are consistent with the studies by Kim et al. and Xiao et al., where PPV5 was detected in fattening pigs with 22.9% (57/249) and 12.3% (29/235) detection frequency, respectively [6,8].

Furthermore, it has to be mentioned that in our study, the number of samples significantly varied each year, as the samples were received as a part of routine diagnostic surveillance, and the data about animal age in companion documents were recorded in a wide range or occasionally missed. Thus, these factors were the principal limitations in establishing the virus prevalence on farms among different age groups and in the country as a whole.

Based on the phylogenetic analysis, Russian PPV5 isolates were genetically similar to the globally identified strains and belonged to three different clades. Isolates from the Tomsk Region were clustered into distinct clades. The Tomsk-74 isolate formed a clade with the isolate from a healthy suckling pig from the USA [24], whereas Tomsk-84 and Burytia-302 had high nucleotide identity with the Colombian isolate identified in a fattening pig [5]. Thus, the Moscow-019 and Moscow-4060 isolates showed high nucleotide identity with the AH-PPV520178-2 strain from China (MW853953). This Chinese strain was detected in coinfection with PCV2 in a piglet with clinical respiratory signs [25]. The previously characterized Russian isolate from the wild boar obtained from the Moscow Region [18] was highly similar to the Moscow isolates; however, it was assigned to a separate branch with the isolate from Columbia. At the same time, the bootstrap support values were quite low (<70%) for most of the nodes that can be explained by a limited amount of PPV5 sequences from different localities, especially from European countries.

Given that PPV5 is a novel discovered virus, increasing investigations have focused on its viral epidemiology and molecular genetic characteristics, and there is lack of data on its replication in vitro. After the identification and sequencing of PPV5 field isolates, the next step of our study was to attempt the virus isolation in a cell culture system. For this purpose, two serum samples from the Moscow and Tomsk Regions (the Moscow-4060 and Tomsk-84 isolates) with the lowest Ct value were selected. It was necessary to determine a susceptible cell culture for PPV5 accumulation for the in vitro study. Primary cell cultures are widely recognized as quite sensitive systems, since they support the replication of the widest range of viruses [26]. According to this fact, PPTCs were derived and infected three hours post-seeding during the early S-phase; as for parvovirus replication, host cellular DNA polymerase is necessary, and actively dividing cells for productive propagation are required [27,28].

It has been established that members of the *Protoparvovirus* genus are capable of causing cell death by inducing either apoptotic or necrotic pathways, and the infection outcome is primarily dependent on the cell type and the viral strain [29,30,31]. The observed morphological changes during the Moscow-4060 strain cultivation were consistent with the previously described reports on PV replication, demonstrating partial monolayer destruction, the formation of multinucleated cells and large cytoplasmic vacuoles, and various alterations in nuclei, including chromatin condensation, apoptotic body formation, and nuclear fragmentation [23,32,33,34]. As both nuclear and cytoplasmic alterations were observed during the study, it is necessary to investigate the PPV5 penetration pathways in cells and its further localization with the application of electron microscopy. Furthermore, we have confirmed the virus replication by qPCR in this case, as the viral DNA was detected in spite of consequent ten-fold dilution during seven passages. By contrast, the evident replicative signs during the Tomsk-84 strain cultivation were not observed due to its probable non-viability or the low viral load. In summary, based on our findings, primary porcine testicular cells might be considered a promising model for the PPV5 isolation and its subsequent propagation.

## 5. Conclusions

In conclusion, our results demonstrated the first evidence of PPV5 circulation in domestic pigs in the territory of Russia. The retrospective analysis revealed that PPV5 has been circulating in swine herds since at least 2021. Further research is necessary to gain a better understanding of PPV5 epidemiology on pig farms in various regions of Russia. PPV5 isolation and its propagation in cell culture is the first step for an in-depth study of the virus in vitro and in vivo. Our results may shed light on the pathogenesis of PPV5 infection and provide insights into the relationships between the host and the virus. Furthermore, the isolation of PPV5 in PPTCs enables the production of the purified virus antigen for obtaining specific antibodies.

## Figures and Tables

**Figure 1 vetsci-12-00535-f001:**
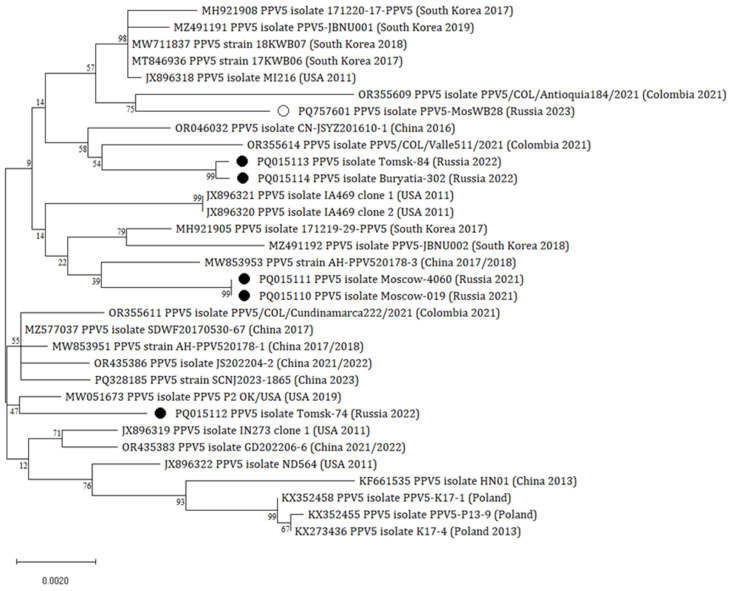
Phylogenetic dendrogram based on the alignment of the capsid protein gene. The isolates obtained in this study are designated by a black circle (●). The isolate obtained from the wild boar in Russia is designated by a white circle (○). The dendrogram was inferred using the ML method and the GTR model (G + I). The robustness of the topology was confirmed by 1000 bootstrap replicates. The scale bar indicates the number of substitutions per site.

**Figure 2 vetsci-12-00535-f002:**
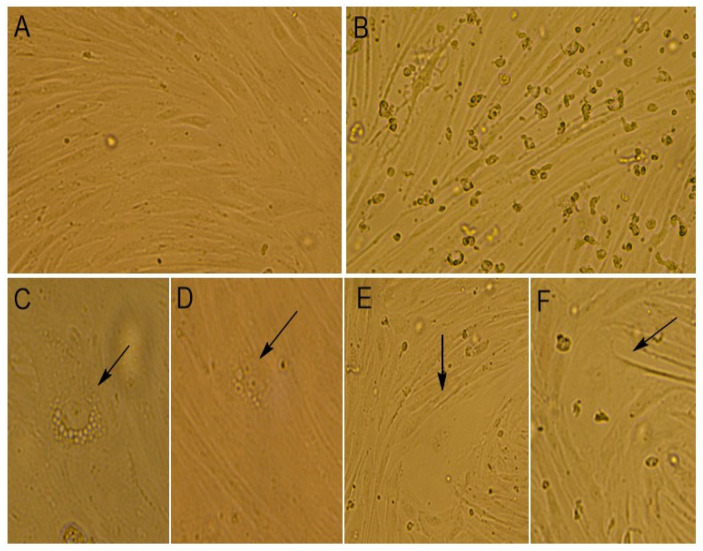
Light microscopy of PPV5-infected and mock-infected PPTCs. Morphological changes observed during 6th passage of Moscow-4060 strain are presented. (**A**) Mock-infected PPTCs, magnification 100×; (**B**) PPV5-infected PPTCs, magnification 100×; (**C**) enlargement of the nucleus, strong vacuolization in the cytoplasm near the nucleus, magnification 400×; (**D**) PPV5-infected cells: giant cell, large vacuolization in the cytoplasm near the nucleus, magnification 400×; (**E**) giant, multinucleated cell, magnification 400×; (**F**) symplast formation, magnification 400×. Arrows indicate specific morphological alterations in PPV5-infected cells.

**Figure 3 vetsci-12-00535-f003:**
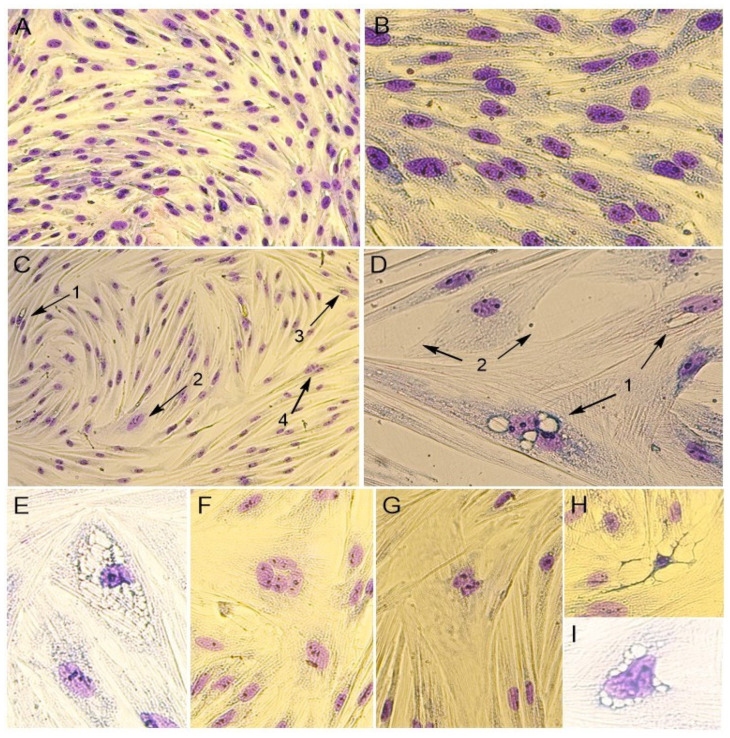
Cytomorphological changes following PPV5 infection. Stained PPV5-infected cells with Moscow-4060 strain at level of 6th passage are presented. (**A**) Mock-infected PPTCs, magnification 100×; (**B**) mock-infected PPTCs, magnification 200×; (**C**) PPV5-infected cells: 1—large vacuole in the cytoplasm; 2—giant cell with enlarged nucleus; 3—vacuole in the cytoplasm near the nucleus; 4—nucleus deformation, magnification 100×; (**D**) PPV5-infected cells: 1—giant cell; large vacuolization in the cytoplasm near the nucleus; 2—unformed monolayer, magnification 630×; (**E**) complete vacuolization of the cytoplasm, nuclear alteration, and nucleoli enlargement, magnification 630×; (**F**) symplast formation, magnification 630×; (**G**) giant, multinucleated cell, magnification 630×; (**H**) nuclear pyknosis, magnification 400×; (**I**) deformed nucleus surrounded by large vacuoles, magnification 630×.

**Table 1 vetsci-12-00535-t001:** Investigated farms and percentage of PPV5-positive samples.

Regions	Farms *	2020	2021	2022	2023	Total Detection Rate
Vologda Region	VR1		0/38 (0%)			0/38 (0%)
Republic of Buryatia	RB1			0/30 (0%)		21/143 (14.7%)
	RB2			5/43 (11.6%)		
	RB3			16/70 (22.9%)		
Moscow Region	MR1		7/42 (16.6%)			7/50 (14.0%)
	MR2			0/8 (0%)		
Kemerovo Region	KR1			4/51 (7.8%)		4/59 (6.8%)
	KR2			0/8 (0%)		
Tomsk Region	TR1			2/95 (2.1%)		11/131 (8.4%)
	TR2			9/36 (25.0%)		
Sverdlovsk Region	SR1			7/44 (15.9%)		7/44 (15.9%)
Krasnoyarsk Krai	KK1			0/17 (0%)		15/139 (10.8%)
	KK2	0/17 (0%)		12/69 (17.4%)		
	KK3			3/10 (30%)		
	KK4	0/18 (0%)		0/8 (0%)		
Pskov Region	PR1				0/90 (0%)	0/138 (0%)
	PR2				0/48 (0%)	
Republic of Mordovia	RM1			0/80 (0%)		9/160 (5.6%)
	RM2			9/80 (11.3%)		
Belgorod Region	BR1			14/82 (17.1%)		14/82 (17.1%)
Total	20	0/35 (0%)	7/80 (8.75%)	81/731 (11.1%)	0/138 (0%)	88/984 (8.9%)

* Capitalized letters and the numbers signify the short name of the region and the farm number (e.g., VR1—the Vologda Region, Farm 1).

**Table 2 vetsci-12-00535-t002:** PPV5 detection rates by age groups.

Age Group *	Total	Positive	Detection Rate (%)
Weaned piglets	94	1	1.1%
Sows	92	12	13.0%
Fattening pigs	330	45	13.6%

* Age groups: sows (aged 2–3 years), fattening pigs (aged 3–9 months), and weaned piglets (aged 20–60 days).

**Table 3 vetsci-12-00535-t003:** The Ct value of PPV5 isolates over seven serial passages.

Isolate	GenBank Accession Number	Region	Initial Sample Ct Value	PPV5 qPCR Results						
				P1	P2	P3	P4	P5	P6	P7
Moscow-4060	PQ015111	Moscow Region	13.9	17.9	17.8	19.5	21.9	22.9	21.8	20.8
Tomsk-84	PQ015113	Tomsk Region	16.3	21.0	23.9	-	nc	nc	nc	nc

“P”—passage of virus; “-”—negative; “nc”—not conducted.

## Data Availability

The nucleotide sequences are available at the NCBI GenBank under accession numbers PQ015110-PQ015114.

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
