# Peer review of "Molecular Detection of Porcine Parvovirus 5 in Domestic Pigs in Russia and Propagation of Field Isolates in Primary Porcine Testicular Cells"

_vetsci, 2025, doi:10.3390/vetsci12060535_

Round 1

Reviewer 1 Report

Comments and Suggestions for Authors

The present work describes the molecular prevalence of PPV-5 in Russia (in particular some regions). The study presents information worthy of publication although some comments need to be addressed first. In general the manuscript has a rather simple language, some sentences (too short) could be merged. My specific comments are as follows: 1) Line 16: Are only 20 farms sufficient to estimate the prevalence at the farm level in Russia?
2) It is unclear the purpose of the in vitro characterization of PPV-5 that was performed and why it was performed after sequencing.
3) Although the results are reported by region and age class, they were not statistically validated (e.g. with chi-square).
4) Line 36: Are the authors able to define apoptotic and necrotic cells by observation alone?
5) There are no conclusions in the abstract.
6) More information on the spread in classic matrices (abortions and reproductive organs for abortions and feces for diarrheal syndromes) in pigs and wild boar should be reported. Below are some studies that could be useful. "Detection of selected pathogens in reproductive tissues of wild boars in the Campania region, southern Italy"; "Detection and molecular characterization of porcine parvovirus in fetal tissues from sows without reproductive failure in Argentina"; "Characterization of porcine parvovirus type 3 and porcine circovirus type 2 in wild boars (Sus scrofa) in Slovakia"
7) Line 82: How was it calculated? How was the number of farms calculated? How many pig farms are there in the study area? Was convenience sampling applied? If so, it is a limitation of the study that should be highlighted.
8) On what basis (age or weight) were the pigs divided?
9) More information about the protocol (kits used, primers, protocol, target gene etc.) would be useful.
10) Lack of statistical analysis in the results. I also recommend adding sex as a variable (if information is available)
11) Line 350: Specify that the results obtained in other species change in the basic matrix, protocol etc.
12) Discuss age and provenance after statistical analysis.

Author Response

Comment 1: The present work describes the molecular prevalence of PPV-5 in Russia (in particular some regions). The study presents information worthy of publication although some comments need to be addressed first. In general the manuscript has a rather simple language, some sentences (too short) could be merged.

Response 1: We greatly appreciate the time and effort you have put into reviewing our manuscript and are grateful for your insightful comments. According to your comment, some sentences were merged to improve the text quality.

Comment 2: My specific comments are as follows: Line 16: Are only 20 farms sufficient to estimate the prevalence at the farm level in Russia?

Response 2: According to your comment, we agree that 20 farms are not sufficient for estimating the prevalence of PPV5 at the level of the country.  In this study, we investigated the detection rate of the virus by testing samples that were received during routine diagnostic surveillance, and the prevalence estimation was not included in the objectives of the study. Consequently, to calculate the prevalence in the country as a whole, more farms and samples from all Russian regions practicing pig farming are necessary to analyze. We had provided this information in the lines 365-370.

Comment 3: It is unclear the purpose of the in vitro characterization of PPV-5 that was performed and why it was performed after sequencing.

Response 3: At the initial stage of our study, we aimed to detect the PPV5 in samples. In turn, sequencing was necessary in order to confirm the virus presence in samples and determine nucleotide sequences. Further, it was of scientific interest to study the possibility of the virus isolation in a cell culture system, as there was a lack in the publishing data. The purpose of the in vitro characterization was to study the replication of PPV5 in infected cells, as well as its accumulation. So, these parts of the study were independent and possessed different objectives.

Comment 4: Although the results are reported by region and age class, they were not statistically validated (e.g. with chi-square).

Response 4: As we have mentioned in the manuscript, we tested the diagnostic samples (lines 84-85) for the presence of PPV5 DNA, and the number of samples varied significantly in each region, so this parameter cannot be proven statistically. According to the age category, we applied exact Fisher test (lines 177-180), and the results were statistically validated (lines 240-244)

Comment 5: Line 36:Are the authors able to define apoptotic and necrotic cells by observation alone?

Response 5: During cultivation, we have noticed the cytopathogenic effect of the virus in PPTCs, characterized by vacuolization of the cell cytoplasm and rounding and death of cells. Therefore, it was described in the manuscript. However, in order to determine and prove the certain apoptotic and necrotic features, flow cytometry and Annexin/ FITC/PI staining can be conducted in the further studies. The corresponding lines was modified according to your comments (lines 29-30).

Comment 6: There are no conclusions in the abstract.

Response 6: The abstract has been updated and revised (lines 18-32)

Comment 7: More information on the spread in classic matrices (abortions and reproductive organs for abortions and feces for diarrheal syndromes) in pigs and wild boar should be reported. Below are some studies that could be useful. "Detection of selected pathogens in reproductive tissues of wild boars in the Campania region, southern Italy"; "Detection and molecular characterization of porcine parvovirus in fetal tissues from sows without reproductive failure in Argentina"; "Characterization of porcine parvovirus type 3 and porcine circovirus type 2 in wild boars (Sus scrofa) in Slovakia"

Response 7: In this study solely serum samples were analyzed and other specimen types were not available, because of this fact we discussed only information to compare detection rates for the same specimen type. Following your comment, we have added some data from other studies (lines 339-342).

Comment 8: How was it calculated? How was the number of farms calculated? How many pig farms are there in the study area? Was convenience sampling applied? If so, it is a limitation of the study that should be highlighted.

Response 8: The study was conducted based on diagnostic samples, and no convenience sampling was applied. This limitation was pointed in the discussion section (lines 365-370)

Comment 9: On what basis (age or weight) were the pigs divided?

Response 9: Age was the basis for division of pigs. This statement was outlined in lines 82-84.

Comment 10: More information about the protocol (kits used, primers, protocol, target gene etc.) would be useful.

Response 10: The following part was modified in accordance with your comment (lines 89-94).

Comment 11: Lack of statistical analysis in the results. I also recommend adding sex as a variable (if information is available).

Response 11: Statistical analysis was performed using the exact Fisher test for several age categories according to the available data (lines 353-364). To our regret, information about sex was not available.

Comment 12: Line 350: Specify that the results obtained in other species change in the basic matrix, protocol etc.

Response 12: In our study, all available specimens were restricted by serum samples and other types were not provided. The information from the publishing data concerning other sample types has been added to the discussion section in order to highlight this point (lines 339-342).

Comment 13: Discuss age and provenance after statistical analysis.

Response 13: The statistical difference for the age categories was calculated, and this issue has been discussed (lines 353-364). The point concerning the provenance cannot be accurately discussed because of the lack of data.

Reviewer 2 Report

Comments and Suggestions for Authors

The manuscript titled "Molecular detection of porcine parvovirus 5 in domestic pigs in Russia and propagation of field isolates in primary porcine testes cells" presents valuable findings on the first detection and isolation of PPV5 in Russia. The study is well-designed and addresses a significant gap in understanding PPV5 epidemiology and in vitro propagation. However, some aspects require clarification and improvement to enhance the manuscript's impact and readability.

1Table 1: Include a footnote defining abbreviations (e.g., VR1, RB1) for clarity.

2Figure 2/3: Label mock-infected and infected panels more prominently. Consider merging panels if space permits to streamline results.

3Abstract: The phrase "first evidence of PPV5 circulation" appears twice; revise for conciseness.

4References: Ensure all citations (e.g., Komina et al. 2025) are in press or published.

5Include a map (Supplementary Figure 1) showing sampling locations to visualize geographic distribution.

6Discuss any co-infections detected during screening (if data exists), as PPV5 may interact with other pathogens.

7Detection Rates: The regional variation in PPV5 prevalence is intriguing. Consider adding a brief discussion on possible factors (e.g., farm density, biosecurity) influencing these differences.

8Phylogenetic Analysis: The tree (Figure 1) is informative, but bootstrap values for key nodes are low (<70%). Acknowledge this limitation and discuss its implications for interpreting evolutionary relationships.

9Virus Isolation: The differential replication of Moscow-4060 vs. Tomsk-84 strains is notable. Elaborate on potential reasons (e.g., viral load in original samples, strain-specific adaptations).

Author Response

We profoundly appreciate your time and efforts to increase the quality of our work. The coherent point-by-point responses are provided below.

Comment 1: Table 1: Include a footnote defining abbreviations (e.g., VR1, RB1) for clarity.

Response 1: According to your suggestion, this point has been clarified in the manuscript (lines 199-200).

Comment 2: Figure 2/3: Label mock-infected and infected panels more prominently. Consider merging panels if space permits to streamline results.

Response 2: Unfortunately, from our perspective, the merging images can be complicated for comprehension. We have tried to clearly outline the results in the text and thoroughly labeled the figures.

Comment 3: Abstract: The phrase "first evidence of PPV5 circulation" appears twice; revise for conciseness.

Response 3: According to your comment, this point has been revised.

Comment 4: References: Ensure all citations (e.g., Komina et al. 2025) are in press or published.

Response 4: All citations and references were accurately checked. This article has been already published (doi:10.1186/s12864-025-11371-w.).

Comment 5: Include a map (Supplementary Figure 1) showing sampling locations to visualize geographic distribution.

Response 5: The Supplementary Figure 1 has been updated in accordance with your suggestion.

Comment 6: Discuss any co-infections detected during screening (if data exists), as PPV5 may interact with other pathogens.

Response 6: To our regret, we cannot provide co-infection statuses due to the fact that the samples have been tested for the detection of antibodies by ELISA during the routine diagnostic surveillance framework. Moreover, the samples were not tested for the presence of other viral genomes.

Comment 7: Detection Rates: The regional variation in PPV5 prevalence is intriguing. Consider adding a brief discussion on possible factors (e.g., farm density, biosecurity) influencing these differences.

Response 7: In this study, we investigated the detection rate of the PPV5 and tested the diagnostic samples (lines 79-80) for the presence of viral DNA, but the number of samples varied significantly in each region, so the prevalence cannot be estimated. At the same time, some points touching an issue on the farm density were represented in the discussion section of the manuscript (lines 349-352).

Comment 8: Phylogenetic Analysis: The tree (Figure 1) is informative, but bootstrap values for key nodes are low (<70%). Acknowledge this limitation and discuss its implications for interpreting evolutionary relationships.

Response 8: Following your recommendation, this limitation was outlined in the manuscript (lines 381-383)

Comment 9: Virus Isolation: The differential replication of Moscow-4060 vs. Tomsk-84 strains is notable. Elaborate on potential reasons (e.g., viral load in original samples, strain-specific adaptations).

Response 9: The following issue had been mentioned in the discussion section. We linked the differential replication between the strains with non-viability of the Tomsk-84 strain in a cell culture (lines 407-409). 

Round 2

Reviewer 1 Report

Comments and Suggestions for Authors

Callithrix penicillata